# TABFACT: A LARGE-SCALE DATASET FOR TABLE-BASED FACT VERIFICATION

**Wenhu Chen, Hongmin Wang, Jianshu Chen, Yunkai Zhang, Hong Wang,**
**Shiyang Li, Xiyou Zhou, William Yang Wang**
University of California, Santa Barbara, CA, USA
Tencent AI Lab, Bellevue, WA, USA
{wenhuchen,hongmin_wang,yunkai_zhang,hongwang600,william}@ucsb.edu
{shiyangli,xiyou}@cs.ucsb.edu jianshuchen@tencent.com

## ABSTRACT

The problem of verifying whether a textual hypothesis holds based on the given evidence, also known as fact verification, plays an important role in the study of natural language understanding and semantic representation. However, existing studies are mainly restricted to dealing with unstructured evidence (e.g., natural language sentences and documents, news, etc), while verification under structured evidence, such as tables, graphs, and databases, remains under-explored. This paper specifically aims to study the fact verification given semi-structured data as evidence. To this end, we construct a large-scale dataset called TabFact with 16k Wikipedia tables as the evidence for 118k human-annotated natural language statements, which are labeled as either ENTAILED or REFUTED. TabFact is challenging since it involves both soft linguistic reasoning and hard symbolic reasoning. To address these reasoning challenges, we design two different models: Table-BERT and Latent Program Algorithm (LPA). Table-BERT leverages the state-of-the-art pre-trained language model to encode the linearized tables and statements into continuous vectors for verification. LPA parses statements into programs and executes them against the tables to obtain the returned binary value for verification. Both methods achieve similar accuracy but still lag far behind human performance. We also perform a comprehensive analysis to demonstrate great future opportunities. The data and code of the dataset are provided in https://github.com/wenhuchen/Table-Fact-Checking.

## 1 INTRODUCTION

Verifying whether a textual hypothesis is entailed or refuted by the given evidence is a fundamental problem in natural language understanding (Katz & Fodor, 1963; Van Benthem et al., 2008). It can benefit many downstream applications like misinformation detection, fake news detection, etc. Recently, the first-ever end-to-end fact-checking system has been designed and proposed in Hassan et al. (2017). The verification problem has been extensively studied under different natural language tasks such as recognizing textual entailment (RTE) (Dagan et al., 2005), natural language inference (NLI) (Bowman et al., 2015), claim verification (Popat et al., 2017; Hanselowski et al., 2018; Thorne et al., 2018) and multimodal language reasoning (NLVR/NLVR2) (Suhr et al., 2017; 2019). RTE and NLI view a premise sentence as the evidence, claim verification views passage collection like Wikipedia[1] as the evidence, NLVR/NLVR2 views images as the evidence. These problems have been previously addressed using a variety of techniques including logic rules, knowledge bases, and neural networks. Recently large-scale pre-trained language models (Devlin et al., 2019; Peters et al., 2018; Yang et al., 2019; Liu et al., 2019) have surged to dominate the other algorithms to approach human performance on several textual entailment tasks (Wang et al., 2018; 2019).

However, existing studies are restricted to dealing with unstructured text as the evidence, which would not generalize to the cases where the evidence has a highly structured format. Since such structured evidence (graphs, tables, or databases) are also ubiquitous in real-world applications like

---

[1] https://www.wikipedia.org/

United States House of Representatives Elections, 1972

| District | Incumbent | Party | Result | Candidates |
|---|---|---|---|---|
| California 3 | John E. Moss | democratic | re-elected | John E. Moss (d) 69.9% John Rakus (r) 30.1% |
| California 5 | Phillip Burton | democratic | re-elected | Phillip Burton (d) 81.8% Edlo E. Powell (r) 18.2% |
| California 8 | George Paul Miller | democratic | lost renomination democratic hold | Pete Stark (d) 52.9% Lew M. Warden , Jr. (r) 47.1% |
| California 14 | Jerome R. Waldie | republican | re-elected | Jerome R. Waldie (d) 77.6% Floyd E. Sims (r) 22.4% |
| California 15 | John J. Mcfall | republican | re-elected | John J. Mcfall (d) unopposed |

| Entailed Statement | Refuted Statement |
|---|---|
| 1. John E. Moss and Phillip Burton are both re-elected in the house of representative election.
2. John J. Mcfall is unopposed during the re-election.
3. There are three different incumbents from democratic. | 1. John E. Moss and George Paul Miller are both re-elected in the house of representative election.
2. John J. Mcfall failed to be re-elected though being unopposed.
3. There are five candidates in total, two of them are democrats and three of them are republicans. |

Figure 1: Examples from the TABFACT dataset. The top table contains the semi-structured knowledge facts with caption "United...". The left and right boxes below provide several entailed and refuted statements. The error parts are highlighted with red font.

database systems, dialog systems, commercial management systems, social networks, etc, we argue that the fact verification under structured evidence forms is an equivalently important yet under-explored problem. Therefore, in this paper, we are specifically interested in studying fact verification with semi-structured Wikipedia tables (Bhagavatula et al., 2013)[2] as evidence owing to its structured and ubiquitous nature (Jauhar et al., 2016; Zhong et al., 2017; Pasupat & Liang, 2015). To this end, we introduce a large-scale dataset called TABFACT, which consists of 118K manually annotated statements with regard to 16K Wikipedia tables, their relations are classified as ENTAILED and REFUTED[3]. The entailed and refuted statements are both annotated by human workers. With some examples in Figure 1, we can clearly observe that unlike the previous verification related problems, TABFACT combines two different forms of reasoning in the statements, (i) *Linguistic Reasoning*: the verification requires semantic-level understanding. For example, "John J. Mcfall failed to be re-elected though being unopposed." requires understanding over the phrase "lost renomination ..." in the table to correctly classify the entailment relation. Unlike the existing QA datasets (Zhong et al., 2017; Pasupat & Liang, 2015), where the linguistic reasoning is dominated by paraphrasing, TABFACT requires more linguistic inference or common sense. (ii) *Symbolic Reasoning*: the verification requires symbolic execution on the table structure. For example, the phrase "There are three Democrats incumbents" requires both condition operation (where condition) and arithmetic operation (count). Unlike question answering, a statement could contain compound facts, all of these facts need to be verified to predict the verdict. For example, the "There are ..." in Figure 1 requires verifying three QA pairs (total count=5, democratic count=2, republic count=3). The two forms of reasoning are interleaved across the statements making it challenging for existing models.

In this paper, we particularly propose two approaches to deal with such mixed-reasoning challenge: (i) *Table-BERT*, this model views the verification task completely as an NLI problem by linearizing a table as a premise sentence $p$, and applies state-of-the-art language understanding pre-trained model to encode both the table and statements $h$ into distributed representation for classification. This model excels at linguistic reasoning like paraphrasing and inference but lacks symbolic reasoning skills. (ii) *Latent Program Algorithm*, this model applies lexical matching to find linked entities and triggers to filter pre-defined APIs (e.g. argmax, argmin, count, etc). We adopt bread-first-search with memorization to construct the potential program candidates, a discriminator is further utilized to select the most "consistent" latent programs. This model excels at the symbolic reasoning aspects by executing database queries, which also provides better interpretability by laying out the decision rationale. We perform extensive experiments to investigate their performances: the best-achieved accuracy of both models are reasonable, but far below human performance. Thus, we believe that the proposed table-based fact verification task can serve as an important new benchmark towards the goal of building powerful AI that can reason over both soft linguistic form and hard symbolic forms. To facilitate future research, we released all the data, code with the intermediate results.

---

[2]In contrast to the database tables, where each column has strong type constraint, the cell records in our semi-structured tables can be string/data/integer/floating/phrase/sentences.

[3]we leave out NEUTRAL due to its low inter-worker agreement, which is easily confused with REFUTED.

## 2 TABLE FACT VERIFICATION DATASET

First, we follow the previous Table-based Q&A datasets (Pasupat & Liang, 2015; Zhong et al., 2017) to extract web tables (Bhagavatula et al., 2013) with captions from WikiTables[4]. Here we filter out overly complicated and huge tables (e.g. multirows, multicolumns, latex symbol) and obtain 18K relatively clean tables with less than 50 rows and 10 columns.

For crowd-sourcing jobs, we follow the human subject research protocols[5] to pay Amazon Mechanical Turk[6] workers from the native English-speaking countries "US, GB, NZ, CA, AU" with approval rates higher than 95% and more than 500 accepted HITs. Following WikiTableQuestion (Pasupat & Liang, 2015), we provide the annotators with the corresponding table captions to help them better understand the background. To ensure the annotation quality, we develop a pipeline of "positive two-channel annotation" → "negative statement rewriting" → "verification", as described below.

### 2.1 POSITIVE TWO-CHANNEL COLLECTION & NEGATIVE REWRITING STRATEGY

To harvest statements of different difficulty levels, we design a two-channel collection process:

**Low-Reward Simple Channel**: the workers are paid 0.45 USD for annotating one Human Intelligent Task (HIT) that requires writing five statements. The workers are encouraged to produce plain statements meeting the requirements: (i) corresponding to a single row/record in the table with unary fact without involving compound logical inference. (ii) mention the cell values without dramatic modification or paraphrasing. The average annotation time of a HIT is 4.2 min.

**High-Reward Complex Channel**: the workers are paid 0.75 USD for annotating a HIT (five statements). They are guided to produce more sophisticated statements to meet the requirements: (i) involving multiple rows in the tables with higher-order semantics like argmax, argmin, count, difference, average, summarize, etc. (ii) rephrase the table records to involve more semantic understanding. The average annotation time of a HIT is 6.8 min. The data obtained from the complex channel are harder in terms of both linguistic and symbolic reasoning, the goal of the two-channel split is to help us understand the proposed models can reach under different levels of difficulty.

As suggested in (Zellers et al., 2018), there might be annotation artifacts and conditional stylistic patterns such as length and word-preference biases, which can allow shallow models (e.g. bag-of-words) to obtain artificially high performance. Therefore, we design a negative rewriting strategy to minimize such linguistic cues or patterns. Instead of letting the annotators write negative statements from scratch, we let them rewrite the collected entailed statements. During the annotation, the workers are explicitly guided to modify the words, phrases or sentence structures but retain the sentence style/length to prevent artificial cues. We disallow naive negations by adding "not, never, etc" to revert the statement polarity in case of obvious linguistic patterns.

### 2.2 QUALITY CONTROL

To control the quality of the annotation process, we review a randomly sampled statement from each HIT to decide whether the whole annotation job should be rejected during the annotation process. Specifically, a HIT must satisfy the following criteria to be accepted: (i) the statements should contain neither typos nor grammatical errors. (ii) the statements do not contain vague claims like might, few, etc. (iii) the claims should be explicitly supported or contradicted by the table without requiring the additional knowledge, no middle ground is permitted. After the data collection, we re-distribute all the annotated samples to further filter erroneous statements, the workers are paid 0.05 USD per statement to decide whether the statement should be rejected. The criteria we apply are similar: no ambiguity, no typos, explicitly supported or contradictory. Through the post-filtering process, roughly 18% entailed and 27% refuted instances are further abandoned due to poor quality.

---

[4]http://websail-fe.cs.northwestern.edu/wikiTables/about/
[5]https://en.wikipedia.org/wiki/Minimum_wage_in_the_United_States
[6]https://www.mturk.com/

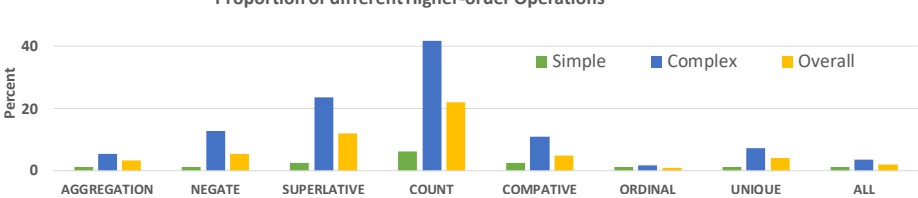

Figure 2: Proportion of different higher-order operations from the simple/complex channels.

| Channel | #Sentence | #Table | Len(Ent) | Len(Ref) | Split | #Sentence | Table | Row | Col |
|---------|-----------|--------|----------|----------|-------|-----------|-------|-----|-----|
| Simple  | 50,244    | 9,189  | 13.2     | 13.1     | Train | 92,283    | 13,182 | 14.1 | 5.5 |
| Complex | 68,031    | 7,392  | 14.2     | 14.2     | Val   | 12,792    | 1,696 | 14.0 | 5.4 |
| Total   | 118,275   | 16,573 | 13.8     | 13.8     | Test  | 12,779    | 1,695 | 14.2 | 5.4 |

Table 1: Basic statistics of the data collected from the simple/complex channel and the division of Train/Val/Test Split in the dataset, where "Len" denotes the averaged sentence length.

## 2.3 DATASET STATISTICS

**Inter-Annotator Agreement**: After the data collection pipeline, we merged the instances from two different channels to obtain a diverse yet clean dataset for table-based fact verification. We sample 1000 annotated (table, statement) pairs and re-distribute each to 5 individual workers to re-label them as either ENTAILED or REFUTED. We follow the previous works (Thorne et al., 2018; Bowman et al., 2015) to adopt the Fleiss Kappa (Fleiss, 1971) as an indicator, where Fleiss $\kappa = \frac{\bar{p}_c - \bar{p}_e}{1 - \bar{p}_e}$ is computed from from the observed agreement $\bar{p}_c$ and the agreement by chance $\bar{p}_e$. We obtain a Fleiss $\kappa = 0.75$, which indicates strong inter-annotator agreement and good-quality.

**Dataset Statistics**: As shown in Table 1, the amount of data harvested via the complex channel slightly outnumbers the simple channel, the averaged length of both the positive and negative samples are indistinguishable. More specifically, to analyze to which extent the higher-order operations are included in two channels, we group the common higher-order operations into 8 different categories. As shown in Figure 2, we sample 200 sentences from two different channels to visualize their distribution. We can see that the complex channel overwhelms the simple channel in terms of the higher-order logic, among which, count and superlatives are the most frequent. We split the whole data roughly with 8:1:1 into train, validation[7], and test splits and shows their statistics in Table 1. Each table with an average of 14 rows and 5-6 columns corresponds to 2-20 different statements, while each cell has an average of 2.1 words. In the training split, the positive instances slightly outnumber the negative instances, while the validation and test split both have rather balanced distributions over positive and negative instances.

## 3 MODELS

With the collected dataset, we now formally define the table-based fact verification task: the dataset is comprised of triple instances $(\mathbf{T}, S, L)$ consisting of a table $\mathbf{T}$, a natural language statement $S = s_1, \cdots, s_n$ and a verification label $L \in \{0, 1\}$. The table $\mathbf{T} = \{T_{i,j} | i \leq R_T, j \leq C_T\}$ has $R_T$ rows and $C_T$ columns with the $T_{ij}$ being the content in the $(i, j)$-th cell. $T_{ij}$ could be a word, a number, a phrase, or even a natural language sentence. The statement S describes a fact to be verified against the content in the table $\mathbf{T}$. If it is entailed by $\mathbf{T}$, then $L = 1$, otherwise the label $L = 0$. Figure 1 shows some entailed and refuted examples. During training, the model and the learning algorithm are presented with $K$ instances like $(\mathbf{T}, S, L)_{k=1}^K$ from the training split. In the testing stage, the model is presented with $(\mathbf{T}, S)_{k=1}^{K'}$ and supposed to predict the label as $\hat{L}$. We measure the performance by the prediction accuracy $Acc = \frac{1}{K'} \sum_1^{K'} \mathbb{I}(\hat{L}_k = L_k)$ on the test set. Before building the model, we first perform entity linking to detect all the entities in the statements. Briefly, we first lemmatize the words and search for the longest sub-string matching pairs between statements and table cells/captions, where the matched phrases are denoted as the linked entities. To focus on statement verification against the table, we do not feed the caption to the model and simply

---

[7]We filter roughly 400 sentences from abnormal tables including hyperlinks, math symbols, etc

mask the phrases in the statements which link to the caption with placeholders. The details of the entity linker are listed in the Appendix. We describe our two proposed models as follows.

## 3.1 LATENT PROGRAM ALGORITHM (LPA)

In this approach, we formulate the table fact verification as a program synthesis problem, where the latent program algorithm is not given in TABFACT. Thus, it can be seen as a weakly supervised learning problem as discussed in Liang et al. (2017); Lao et al. (2011). Under such a setting, we propose to break down the verification into two stages: (i) latent program search, (ii) discriminator ranking. In the first program synthesis step, we aim to parse the statement into programs to represent its semantics. We define the plausible API set to include roughly 50 different functions like *min, max, count, average, filter, and* and realize their interpreter with Python-Pandas. Each API is defined to take arguments of specific types (*number, string, bool, and view (e.g sub-table)*) to output specific-type variables. During the program execution, we store the generated intermediate variables to different-typed caches $\mathcal{N}, \mathcal{R}, \mathcal{B}, \mathcal{V}$ (Num, Str, Bool, View). At each execution step, the program can fetch the intermediate variable from the caches to achieve semantic compositionality. In order to shrink the search space, we follow NSM (Liang et al., 2017) to use trigger words to prune the API set and accelerate the search speed. The definitions of all API, trigger words can be found in the Appendix. The comprehensive the latent program search procedure is summarized in Algorithm 1,

---

**Algorithm 1** Latent Program Search with Comments

1: Initialize Number Cache $\mathcal{N}$, String Cache $\mathcal{R}$, Bool Cache $\mathcal{B}$, View Cache $\mathcal{V} \to \emptyset$
2: Push linked numbers, strings from the given statement $S$ into $\mathcal{N}, \mathcal{R}$, and push $\mathbf{T}$ into $\mathcal{V}$
3: Initialize the result collector $\mathcal{P} \to \emptyset$ and an empty program trace $P = \emptyset$
4: Initialize the Queue $\mathcal{Q} = [(P, \mathcal{N}, \mathcal{R}, \mathcal{B}, \mathcal{V})]$, we use $\mathcal{Q}$ to store the intermediate states
5: Use trigger words to find plausible function set $\mathcal{F}$, for example, $more$ will trigger $Greater$ function.
6: **while** loop over time $t = 1 \to$ MAXSTEP **do**:
7:     **while** $(P, \mathcal{N}, \mathcal{R}, \mathcal{B}, \mathcal{V}) = \mathcal{Q}.pop()$ **do**:
8:         **while** loop over function set $f \in \mathcal{F}$ **do**:
9:             **if** arguments of $f$ are in the caches **then**
10:                 Pop out the required arguments $arg_1, arg_2, \cdots, arg_n$ for different cachess.
11:                 Execute $A = f(arg_1, \cdots, arg_n)$ and concatenate the program trace $P$.
12:                 **if** Type(A)=Bool **then**
13:                     **if** $\mathcal{N} = \mathcal{S} = \mathcal{B} = \emptyset$ **then**
14:                         $\mathcal{P}.push((P, A))$ # The program $P$ is valid since it consumes all the variables.
15:                         $P = \emptyset$ # Collect the valid program $P$ into set $\mathcal{P}$ and reset $P$
16:                     **else**
17:                         $\mathcal{B}.push(A)$ # The intermediate boolean value is added to the bool cache
18:                         $\mathcal{Q}.push((P, \mathcal{N}, \mathcal{R}, \mathcal{B}, \mathcal{V}))$ # Add the refreshed state to the queue again
19:                 **if** Type(A) $\in$ {Num, Str, View} **then**
20:                     **if** $\mathcal{N} = \mathcal{S} = \mathcal{B} = \emptyset$ **then**
21:                         $P = \emptyset$;break # The program ends without consuming the cache, throw it.
22:                     **else**
23:                         push $A$ into $\mathcal{N}$ or $\mathcal{S}$ or $\mathcal{V}$ # Add the refreshed state to the queue for further search
24:                         $\mathcal{Q}.push((P, \mathcal{N}, \mathcal{R}, \mathcal{B}, \mathcal{V}))$
25: Return the triple $(\mathbf{T}, S, \mathcal{P})$ # Return (Table, Statement, Program Set)

---

and the searching procedure is illustrated in Figure 3.

After we collected all the potential program candidates $\mathcal{P} = \{(P_1, A_1), \cdots, (P_n, A_n)\}$ for a given statement $S$ (where $(P_i, A_i)$ refers to $i$-th candidate), we need to learn a discriminator to identify the "appropriate" traces from the set from many erroneous and spurious traces. Since we do not have the ground truth label about such discriminator, we use a weakly supervised training algorithm by viewing all the label-consistent programs as positive instances $\{P_i | (P_i, A_i); A_i = L\}$ and the label-inconsistent program as negative instances $\{P_i | (P_i, A_i); A_i \neq L\}$ to minimize the cross-entropy of discriminator $p_\theta(S, P)$ with the weakly supervised label. Specifically, we build our discriminator with a Transformer-based two-way encoder (Vaswani et al., 2017), where the statement encoder encodes the input statement $S$ as a vector $Enc^S(S) \in \mathbb{R}^{n \times D}$ with dimension $D$, while the program encoder encodes the program $P = p_1, \cdots, p_m$ as another vector $Enc^P(P) \in \mathbb{R}^{m \times D}$, we concatenate these two vectors and feed it into a linear projection layer

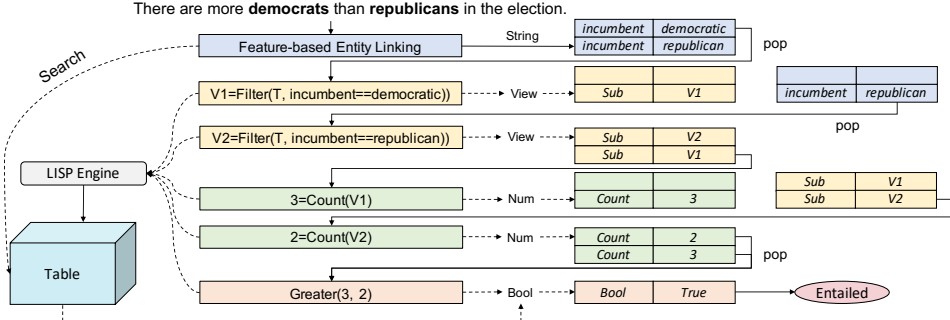

Figure 3: The program synthesis procedure for the table in Figure 1. We link the entity (e.g. *democratic*, *republican*), and then composite functions on the fly to return the values from the table.

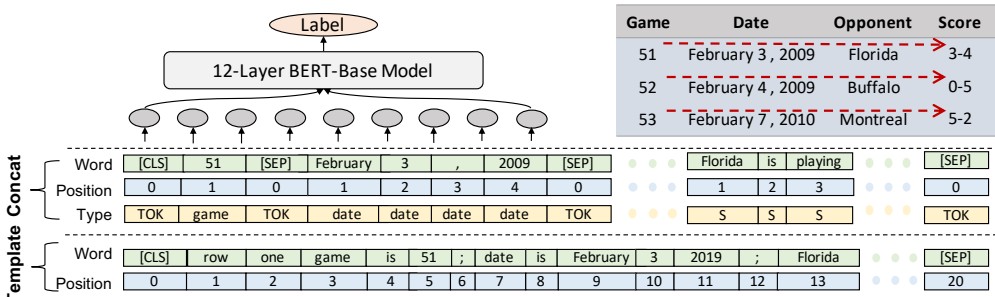

Figure 4: The diagram of Table-BERT with horizontal scan, two different linearizations are depicted.

to compute $p_\theta(S, P) = \sigma(v_p^T[Enc^S(S); Enc^P(P)])$ as the relevance between S and $P$ with weight $v_p \in \mathbb{R}^D$. At test time, we use the discriminator $p_\theta$ to assign confidence $p_\theta(S, P)$ to each candidate $P \in \mathcal{P}$, and then either aggregate the prediction from all hypothesis with the confidence weights or rank the highest-confident hypothesis and use their outputs as the prediction.

## 3.2 TABLE-BERT

In this approach, we view the table verification problem as a two-sequence binary classification problem like NLI or MPRC (Wang et al., 2018) by linearizing a table **T** into a sequence and treating the statement as another sequence. Since the linearized table can be extremely long surpassing the limit of sequence models like LSTM, Transformers, etc. We propose to shrink the sequence by only retaining the columns containing entities linked to the statement to alleviate such a memory issue. In order to encode such sub-table as a sequence, we propose two different linearization methods, as is depicted in Figure 4. (i) Concatenation: we simply concatenate the table cells with [SEP] tokens in between and restart position counter at the cell boundaries; the column name is fed as another type embedding to the input layer. Such design retains the table information in its machine format. (ii) Template: we adopt simple natural language templates to transform a table into a "somewhat natural" sentence. Taking the horizontal scan as an example, we linearize a table as "row one's game is 51; the date is February; ..., the score is 3.4 (ot). row 2 is ...". The isolated cells are connected with punctuations and copula verbs in a language-like format.

After obtaining the linearized sub-table $\tilde{\mathbf{T}}$, we concatenate it with the natural language statement S and prefix a [CLS] token to the sentence to obtain the sequence-level representation $H = f_{BERT}([\tilde{\mathbf{T}}, S])$, with $H \in \mathbb{R}^{768}$ from pre-trained BERT (Devlin et al., 2019). The representation is further fed into multi-layer perceptron $f_{MLP}$ to obtain the entailment probability $p_\theta(\tilde{\mathbf{T}}, S) = \sigma(f_{MLP}(H))$, where $\sigma$ is the sigmoid function. We finetune the model $\theta$ (including the parameters of BERT and MLP) to minimize the binary cross entropy $\mathcal{L}(p_\theta(\tilde{\mathbf{T}}, S), L)$ on the training set. At test time, we use the trained BERT model to compute the matching probability between the (table, statement) pair, and classify it as ENTAILED statement when $p_\theta(\tilde{\mathbf{T}}, S)$ is greater than 0.5.

# 4 EXPERIMENTS

In this section, we aim to evaluate the proposed methods on TABFACT. Besides the standard validation and test sets, we also split the test set into a simple and a complex partition based on the channel from which they were collected. This facilitates analyzing how well the model performs under different levels of difficulty. Additionally, we also hold out a small test set with 2K samples for human evaluation, where we distribute each (table, statement) pair to 5 different workers to approximate human judgments based on their majority voting, the results are reported in Table 2.

| Model | Val | Test | Test (simple) | Test (complex) | Small Test |
|---|---|---|---|---|---|
| BERT classifier w/o Table | 50.9 | 50.5 | 51.0 | 50.1 | 50.4 |
| Table-BERT-Horizontal-F+T-Concatenate | 50.7 | 50.4 | 50.8 | 50.0 | 50.3 |
| Table-BERT-Vertical-F+T-Template | 56.7 | 56.2 | 59.8 | 55.0 | 56.2 |
| Table-BERT-Vertical-T+F-Template | 56.7 | 57.0 | 60.6 | 54.3 | 55.5 |
| Table-BERT-Horizontal-F+T-Template | 66.0 | 65.1 | 79.0 | 58.1 | 67.9 |
| Table-BERT-Horizontal-T+F-Template | **66.1** | 65.1 | **79.1** | **58.2** | **68.1** |
| NSM w/ RL (Binary Reward) | 54.1 | 54.1 | 55.4 | 53.1 | 55.8 |
| NSM w/ LPA-guided ML + RL | 63.2 | 63.5 | 77.4 | 56.1 | 66.9 |
| LPA-Voting w/o Discriminator | 57.7 | 58.2 | 68.5 | 53.2 | 61.5 |
| LPA-Weighted-Voting | 62.5 | 63.1 | 74.6 | 57.3 | 66.8 |
| LPA-Ranking w/ Discriminator | **65.2** | 65.0 | 78.4 | **58.5** | 68.6 |
| LPA-Ranking w/ Discriminator (Caption) | 65.1 | **65.3** | **78.7** | **58.5** | **68.9** |
| Human Performance | - | - | - | - | **92.1** |

Table 2: The results of different models, the numbers are in percentage. T+F means table followed by fact, while F+T means fact followed by table. NSM is modified from Liang et al. (2017).

**NSM** We follow Liang et al. (2017) to modify their approach to fit the setting of TABFACT. Specifically, we adopt an LSTM as an encoder and another LSTM with copy mechanism as a decoder to synthesize the program. However, without any ground truth annotation for the intermediate programs, directly training with reinforcement learning is difficult as the binary reward is under-specified, which is listed in Table 2 as "NSM w/ RL". Further, we use LPA as a teacher to search the top programs for the NSM to bootstrap and then use reinforcement learning to finetune the model, which achieves reasonable performance on our dataset listed as "NSM w/ ML + RL".

**Table-BERT** We build Table-BERT based on the open-source implementation of BERT[8] using the pre-trained model with 12-layer, 768-hidden, 12-heads, and 110M parameters trained in 104 languages. We use the standard BERT tokenizer to break the words in both statements and tables into subwords and join the two sequences with a [SEP] token in between. The representation corresponding to [CLS] is fed into an MLP layer to predict the verification label. We finetune the model on a single TITAN X GPU with a mini-batch size of 6. The best performance is reached after about 3 hours of training (around 10K steps). We implement and compare the following variants of the Table-BERT model including (i) Concatenation vs. Template: whether to use natural language templates during linearization. (ii) Horizontal vs. Vertical: scan direction in linearization.

**LPA** We run the latent program search in a distributed fashion on three 64-core machines to generate the latent programs. The search terminates once the buffer has more than 50 traces or the path length is larger than 7. The average search time for each statement is about 2.5s. For the discriminator model, we design two transformer-based encoders (3 layers, 128-dimension hidden embedding, and 4 heads at each layer) to encode the programs and statements, respectively. The variants of LPA models considered include (i) Voting: assign each program with equal weight and vote without the learned discriminator. (ii) Weighted-Voting: compute a weighted-sum to aggregate the predictions of all latent programs with the discriminator confidence as the weights. (iii) Ranking: rank all the hypotheses by the discriminator confidence and use the top-rated hypothesis as the output. (Caption) means feeding the caption as a sequence of words to the discriminator during ranking.

**Preliminary Evaluation** In order to test whether our negative rewriting strategy eliminates the artifacts or shallow cues, we also fine-tune a pre-trained BERT (Devlin et al., 2019) to classify the statement $S$ without feeding in table information. The result is reported as "BERT classifier w/o

---

[8] https://github.com/huggingface/pytorch-pretrained-BERT

Table" in Table 2, which is approximately the majority guess and reflects the effectiveness of the rewriting strategy. Before presenting the experiment results, we first perform a preliminary study to evaluate how well the entity linking system, program search, and the statement-program discriminator perform. Since we do not have the ground truth labels for these models, we randomly sample 100 samples from the dev set to perform the human study. For the entity linking, we evaluate its accuracy as the number of correctly linked sentences / total sentences. For the latent program search, we evaluate whether the "true" programs are included in the candidate set $\mathcal{P}$ as recall score.

**Results**    We report the performance of different methods as well as human performance in Table 2. First of all, we observe that the naive serialized model fails to learn anything effective (same as the Majority Guess). It reveals the importance of template when using the pre-trained BERT (Devlin et al., 2019) model: the "natural" connection words between individual cells is able to unleash the power of the large pre-trained language model and enable it to perform reasoning on the structured table form. Such behavior is understandable given the fact that BERT is pre-trained on purely natural language corpora. In addition, we also observe that the horizontal scan excels in the vertical scan because it better captures the convention of human expression. Among different LPA methods, we found that LPA-Ranking performs the best since it can better suppress the spurious programs than the voting-based algorithm. Overall, the LPA model is on par with Table-BERT on both simple and test split without any pre-training on external corpus, which reflects the effectiveness of LPA to leverage symbolic operations in the verification process.

Through our human evaluation, we found that only 58% of sentences have been correctly linked without missing-link or over-link, while the systematic search has a recall of 51% under the cases where the sentence is correctly linked. With that being said, the chance for LPA method to cover the correct program (rationale) is roughly under 30%. After the discriminator's re-ranking step, the probability of selecting these particular oracle program is even much lower. However, we still observe a final overall accuracy of 65%, which indicates that the spurious problem is quite severe in LPA, where the correct label is predicted based on the wrong reason.

Through our human evaluation, we also observe that Table-BERT exhibits poor consistency as it can misclassify simple cases but correctly-classify hard cases. These two major weaknesses are yet to be solved in future studies. In contrast, LPA behaves much more consistently and provides a clear latent rationale for its decision. But, such a pipeline system requires laborious handcrafting of API operations and is also very sensitive to the entity linking accuracy. Both methods have pros and cons; how to combine them still remains an open question.

**Program Annotation**    To further promote the development of different models in our dataset, we collect roughly 1400 human-annotated programs paired with the original statements. These statements include the most popular logical operations like superlative, counting, comparison, unique, etc. We provide these annotations in Github[9], which can either be used to bootstrap the semantic parsers or provide the rationale for NLI models.

## 5  RELATED WORK

**Natural Language Inference & Reasoning:** Modeling reasoning and inference in human language is a fundamental and challenging problem towards true natural language understanding. There has been extensive research on RTE in the early years (Dagan et al., 2005) and more recently shifted to NLI (Bowman et al., 2015; Williams et al., 2017). NLI seeks to determine whether a natural language hypothesis $h$ can be inferred from a natural language premise $p$. With the surge of deep learning, there have been many powerful algorithms like the Decomposed Model (Parikh et al., 2016), Enhanced-LSTM (Chen et al., 2017) and BERT (Devlin et al., 2019). Besides the textual evidence, NLVR (Suhr et al., 2017) and NLVR2 (Suhr et al., 2019) have been proposed to use images as the evidence for statement verification on multi-modal setting. Our proposed fact verification task is closely related to these inference tasks, where our semi-structured table can be seen as a collection of "premises" exhibited in a semi-structured format. Our proposed problem hence could be viewed as the generalization of NLI under the semi-structured domain.

**Table Question Answering:** Another line of research closely related to our task is the table-based

---

[9]https://github.com/wenhuchen/Table-Fact-Checking/tree/master/bootstrap

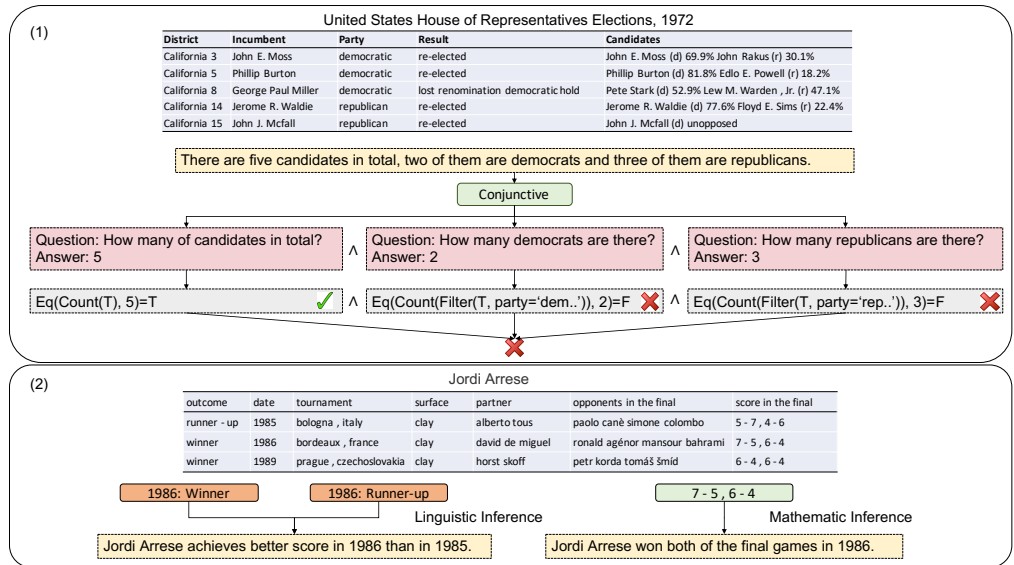

Figure 5: The two uniqueness of Table-based fact verification against standard QA problems.

question answering, such as MCQ (Jauhar et al., 2016), WikiTableQuestion (Pasupat & Liang, 2015), Spider (Yu et al., 2018), Sequential Q&A (Iyyer et al., 2017), and WikiSQL (Zhong et al., 2017), for which approaches have been extended to handle large-scale tables from Wikipedia (Bhagavatula et al., 2013). However, in these Q&A tasks, the question types typically provide strong signals needed for identifying the type of answers, while TABFACT does not provide such specificity. The uniqueness of TABFACT lies in two folds: 1) a given fact is regarded as a false claim as long as any part of the statement contains misinformation. Due to the conjunctive nature of verification, a fact needs to be broken down into several sub-clauses or (Q, A) pairs to separate evaluate their correctness. Such a compositional nature of the verification problem makes it more challenging than a standard QA setting. On one hand, the model needs to recognize the multiple QA pairs and their relationship. On the other hand, the multiple sub-clauses make the semantic form longer and logic inference harder than the standard QA setting. 2) some facts cannot even be handled using semantic forms, as they are driven by linguistic inference or common sense. In order to verify these statements, more inference techniques have to be leveraged to enable robust verification. We visualize the above two characteristics of TABFACT in Figure 5.

**Program Synthesis & Semantic Parsing:** There have also been great interests in using program synthesis or logic forms to solve different natural language processing problems like question answering (Liang et al., 2013; Berant et al., 2013; Berant & Liang, 2014), visual navigation (Artzi et al., 2014; Artzi & Zettlemoyer, 2013), code generation (Yin & Neubig, 2017; Dong & Lapata, 2016), SQL synthesis (Yu et al., 2018), etc. The traditional semantic parsing papers (Artzi et al., 2014; Artzi & Zettlemoyer, 2013; Zettlemoyer & Collins, 2005; Liang et al., 2013; Berant et al., 2013) greatly rely on rules, lexicon to parse natural language sentences into different forms like lambda calculus, DCS, etc. More recently, researchers strive to propose neural models to directly perform end-to-end formal reasoning like Theory Prover (Riedel et al., 2017; Rocktäschel & Riedel, 2017), Neural Turing Machine (Graves et al., 2014), Neural Programmer (Neelakantan et al., 2016; 2017) and Neural-Symbolic Machines (Liang et al., 2017; 2018; Agarwal et al., 2019). The proposed TABFACT serves as a great benchmark to evaluate the reasoning ability of different neural reasoning models. Specifically, TABFACT poses the following challenges: 1) spurious programs (i.e., wrong programs with the true returned answers): since the program output is only a binary label, which can cause serious spurious problems and misguide the reinforcement learning with the under-specified binary rewards. 2) decomposition: the model needs to decompose the statement into sub-clauses and verify the sub-clauses one by one, which normally requires the longer logic inference chains to infer the statement verdict. 3) linguistic reasoning like inference and paraphrasing.

**Fact Checking** The problem of verifying claims and hypotheses on the web has drawn significant attention recently due to its high social influence. Different fact-checking pioneering studies have been

performed including LIAR (Wang, 2017), PolitiFact (Vlachos & Riedel, 2014), FEVER (Thorne et al., 2018) and AggChecker (Jo et al., 2019), etc. The former three studies are mainly based on textual evidence on social media or Wikipedia, while AggChecker is closest to ours in using relational databases as the evidence. Compared to AggChecker, our paper proposes a much larger dataset to benchmark the progress in this direction.

## 6 CONCLUSION

This paper investigates a very important yet previously under-explored research problem: semi-structured fact verification. We construct a large-scale dataset and proposed two methods, Table-BERT and LPA, based on the state-of-the-art pre-trained natural language inference model and program synthesis. In the future, we plan to push forward this research direction by inspiring more sophisticated architectures that can perform both linguistic and symbolic reasoning.

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

# A APPENDIX

## A.1 FUNCTION DESCRIPTION

We list the detailed function description in Figure 6. We also visualize the functionality of the most

| Name | Arguments | Output | Comment |
|---|---|---|---|
| Count | View | Number | Return the number of rows in the View |
| Within | View, Header String, Cell String/Number | Bool | Return whether the cell string/number exists under the Header Column of the given view |
| Without | View, Header String, Cell String/Number | Bool | Return whether the cell string/number does not exist under the Header Column of the given view |
| None | String | Bool | Whether the string represents None, like "None", "No", "-", "No information provided" |
| Before/After | Row, Row | Row | Returns whether row1 is before/after row2 |
| First/Second/Third/Fourth | View, Row | Bool | Returns whether the row is in the first/second/third position of the view |
| Average/Sum/Max/Min | View, Header String | Number | Returns the average/summation/max/min value under the Header Column of the given view |
| Argmin/ Argmax | View, Header String | Row | Returns the row with the maximum/minimum value under the Header Column of the given view |
| Hop | Row, Header String | Number/ String | Returns the cell value under the Header Column of the given row |
| Diff/Add | Number, Number | Number | Perform arithmetic operations on two numbers |
| Greater/Less | Number, Number | Bool | Returns whether the first number is greater/less than the second number |
| Equal/ Unequal | String, String/ Number, Number | Bool | Compare two numbers or strings to see whether they are the same |
| Filter_eq/ Filter_greater/ Filter_less/ Filter_greater_or_equal/ Filter_less_or_equal | View, Header String, Number | View | Returns the subview of the given with the cell values under the Header column greater/less/eq/... against the given number |
| All_eq/All_greater/ All_less/All_greater_or_equa l/All_less_or_equal | View, Header String, Number | Bool | Returns the whether all of the cell values under the Header column are greater/less/eq/... against the given number |
| And/Or | Bool, Bool | Bool | Returns the Boolean operation results of two inputs |

Figure 6: The function definition used in TabFact.

typical functions and their input/output examples in Figure 7.

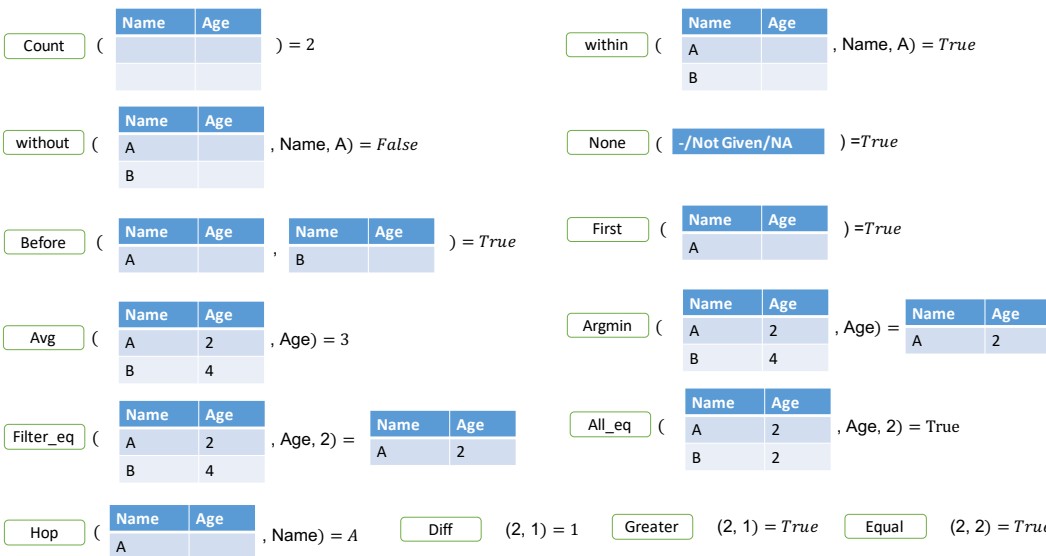

Figure 7: The visualization of different functions.

We list all the trigger words for different functions in Figure 8

| Trigger | Function |
|---|---|
| 'average' | average |
| 'difference', 'gap', 'than', 'separate' | diff |
| 'sum', 'summation', 'combine', 'combined', 'total', 'add', 'all', 'there are' | ddd, sum |
| 'not', 'no', 'never', "didn't", "won't", "wasn't", "isn't","haven't", "weren't", "won't", 'neither', 'none', 'unable', 'fail', 'different', 'outside', 'unable', 'fail' | not_eq, not_within, Filter_not_eq, none |
| 'not', 'no', 'none' | none |
| 'first', 'top', 'latest', 'most' | first |
| 'last', 'bottom', 'latest', 'most' | last |
| 'RBR', 'JJR', 'more', 'than', 'above', 'after' | filter_greater, greater |
| 'RBR', 'JJR', 'less', 'than', 'below', 'under' | filter_less, less |
| 'all', 'every', 'each' | all_eq, all_less, all_greater, |
| ['all', 'every', 'each'], ['not', 'no', 'never', "didn't", "won't", "wasn't"] | all_not_eq |
| 'at most', 'than' | all_less_eq, all_greater_eq |
| 'RBR', 'RBS', 'JJR', 'JJS' | max, min |
| 'JJR', 'JJS', 'RBR', 'RBS', 'top', 'first' | argmax, argmin |
| 'within', 'one', 'of', 'among' | within |
| 'follow', 'following', 'followed', 'after', 'before', 'above', 'precede' | before |
| 'follow', 'following', 'followed', 'after', 'before', 'above', 'precede' | after |
| 'most' | most_freq |
| ordinal | First, second, third, fourth |

Figure 8: The trigger words used to shrink the search space.

## B  HIGHER-ORDER OPERATIONS

1. Aggregation: the aggregation operation refers to sentences like "the averaged age of all ....", "the total amount of scores obtained in ...", etc.

2. Negation: the negation operation refers to sentences like "xxx did not get the best score", "xxx has never obtained a score higher than 5".

3. Superlative: the superlative operation refers to sentences like "xxx achieves the highest score in", "xxx is the lowest player in the team".

4. Comparative: the comparative operation refers to sentences like "xxx has a higher score than yyy".

5. Ordinal: the ordinal operation refers to sentences like "the first country to achieve xxx is xxx", "xxx is the second oldest person in the country".

6. Unique: the unique operation refers to sentences like "there are 5 different nations in the tournament, ", "there are no two different players from U.S"

7. All: the for all operation refers to sentences like "all of the trains are departing in the morning", "none of the people are older than 25."

8. None: the sentences which do not involve higher-order operations like "xxx achieves 2 points in xxx game", "xxx player is from xxx country".

## C  ERROR ANALYSIS

Before we quantitatively demonstrate the error analysis of the two methods, we first theoretically analyze the bottlenecks of the two methods as follows:

**Symbolic**  We first provide a case in which the symbolic execution can not deal with theoretically in Figure 9. The failure cases of symbolic are either due to the entity link problem or function coverage problem. For example, in the given statement below, there is no explicit mention of "7-5, 6-4" cell. Therefore, the entity linking model fails to link to this cell content. Furthermore, even

though we can successfully link to this string, there is no defined function to parse "7-5, 6-5" as "won two games" because it requires linguistic/mathematical inference to understand the implication from the string. Such cases are the weakness of symbolic reasoning models.

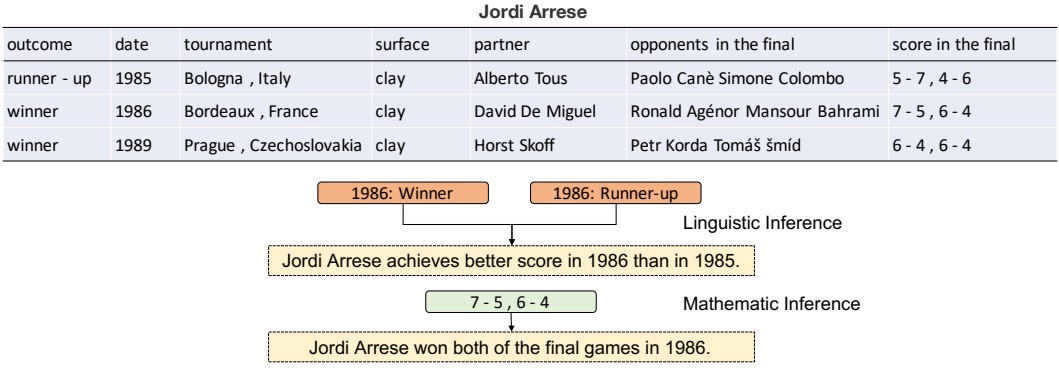

Figure 9: The error case of symbolic reasoning model

**BERT** In contrast, Table-BERT model seems to have no coverage problem as long as it can feed the whole table content. However, due to the template linearization, the table is unfolded into a long sequence as depicted in Figure 10. The useful information, "clay" are separated in a very long span of unrelated words. How to grasp such a long dependency and memorize the history information poses a great challenge to the Table-BERT model.

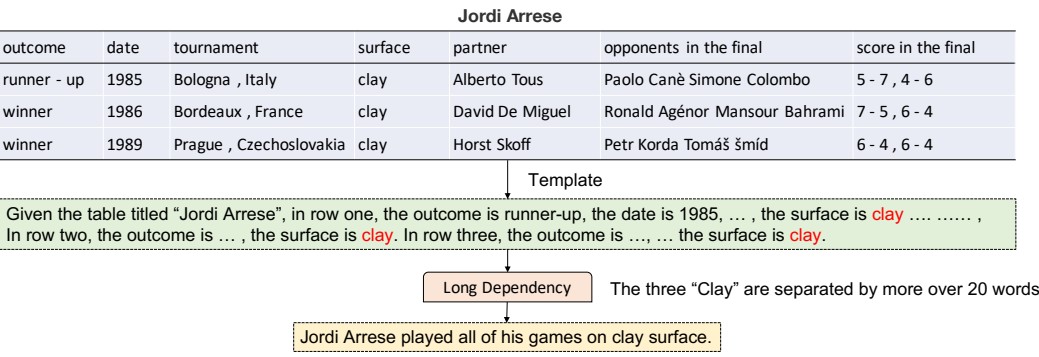

Figure 10: The error case of BERT NLI model

**Statistics** Here we pick 200 samples from the validation set which only involve single semantic and divide them into different categories. We denote the above-mentioned cases as "linguistic inference", and the sentences which only describe information from one row as "Trivial", the rest are based on their logic operation like Aggregation, Superlative, Count, etc. We visualize the accuracy of LPA and Table-BERT in Figure 11. From which we can observe that the statements with linguistic inference are much better handled with the BERT model, while LPA achieves an accuracy barely higher than a random guess. The BERT model can deal with trivial cases well as it uses a horizontal scan order. In contrast, the LPA model outperforms BERT on higher-order logic cases, especially when the statement involves operations like Count and Superlative.

### Error Analysis of LPA/Table-BERT

Figure 11: The error analysis of two different models

## D    REASONING DEPTH

Given that our LPA has the breadth to cover a large semantic space. Here we also show the reasoning depth in terms of how many logic inference steps are required to tackle verify the given claims. We visualize the histogram in Figure 12 and observe that the reasoning steps are concentrated between 4 to 7. Such statistics indicate the difficulty of fact verification in our TABFACT dataset.

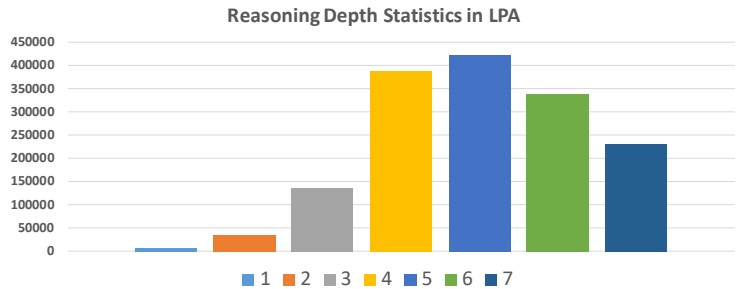

Figure 12: The histogram of reasoning steps required to verify the claims

## E    WHETHER TO KEEP WIKIPEDIA CONTEXT

Before crowd-sourcing the annotation for the tables, we observed that the previous WikiTableQuestion Pasupat & Liang (2015) provides context (Wikipedia title) during annotation while the WikiSQL Zhong et al. (2017) does not. Therefore, we particularly design ablation annotation tasks to compare the annotation quality between w/ and w/o Wikipedia title as context. We demonstrate a typical example in Figure 13, where a Wiki table[10] aims to describe the achievements of a tennis player named Dennis, but itself does not provide any explicit hint about "Tennis Player Dennis". Unsurprisingly, the sentence fluency and coherence significantly drop without such information. Actually, a great portion of these Wikipedia tables requires background knowledge (like sports, celebrity, music, etc) to understand. We perform a small user study to measure the fluency of annotated statements. Specifically, we collected 50 sentences from both annotation w/ and w/o title context and randomly shuffle them as pairs, which are distributed to the 8 experts without telling them their source to compare the language fluency. It turns out that the experts ubiquitously agree that the statements with Wikipedia titles are more human-readable. Therefore, we argue that such a context is necessary for annotators to understand the background knowledge to write more fluent sentences. On the other end, we also hope to minimize the influence of the textual context in the table-based verification task, therefore, we design an annotation criterion: the Wikipedia title

[10]https://en.wikipedia.org/wiki/Dennis_Ralston

is provided to the workers during the annotation, but they are explicitly banned from bringing any unrelated background information other than the title into the annotation. As illustrated in Figure 13, the title only acts as a placeholder in the statements to make it sound more natural.

| outcome | year | championship | surface | partner |
|---------|------|--------------|---------|---------|
| winner | 1960 | Wimbledon championships | grass | Rafael Osuna |
| winner | 1961 | US Championships | grass | Chuck Mckinley |
| runner - up | 1962 | US Championships | grass | Chuck Mckinley |
| winner | 1963 | US Championships (2) | grass | Chuck Mckinley |
| Context (Title) | *Richard Dennis Ralston (born July 27, 1942, an American former tennis player* | | *No Information is provided* | |
| Annotate | From 1960 to 1969, Ralston won five major double championships. | | Winner is on the grass surface. Rafael Osuna is partner in the Wimbeldon | |

Figure 13: Comparison of worker annotation w/ and w/o Wikipedia title as context

## F   ENTITY LINKING

Here we propose to use the longest string match to find all the candidate entities in the table, when multiple candidates coexist, we select the one with the minimum edit distances. The visualization is demonstrated in Figure 14.

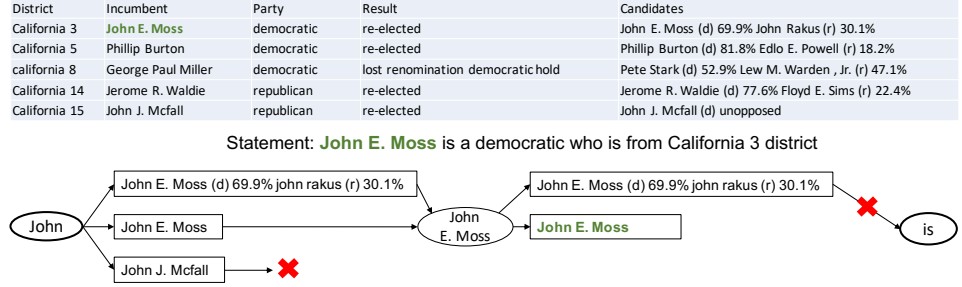

Figure 14: Entity Linking System.

## G   THE PROGRAM CANDIDATES

Here we demonstrate some program candidates in Figure 15, and show how our proposed discriminator is designed to compute the matching probability between the statement and program. Specifically, we employ two transformer-based encoder Vaswani et al. (2017), the left one is aimed to encode the program sequence and the right one is aimed to encode the statement sequence. Their output from [CLS] position is concatenated and fed into an MLP to classify the verification label.

## H   HIT INTERFACE

We provide the human intelligent task interface on AMT in the following. Very detailed instructions on what are trivial statements and what are non-trivial statements. Comprehensive examples have been given to guide the Turkers to write well-formed while logically plausible statements. In order to harvest fake statements without statistical cues, we also provide detailed instructions on how to re-write the "fake" statements. During the annotation, we hire 8 experts to perform sanity checks on each of the HIT to make sure that the annotated dataset is clean and meets our requirements.

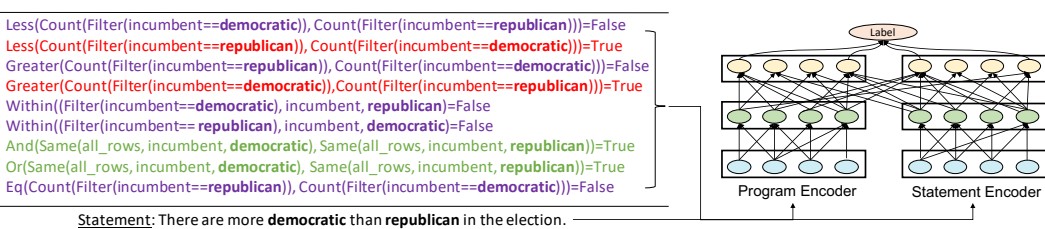

Figure 15: We demonstrate the top program candidates and use the discriminator to rank them.

---

**Survey Instructions** (Click to expand)

**You are given a table with its wikipedia source, your job is to compose non-trivial statements supported by the table.**

- **"Trivial"**: the sentence can be easily generated by looking **only a certain row** without understanding the table.
- **"Non-trivial"**: the sentence requires reading multiple rows of the table and understanding of the table content. For example, the sentences which include **summarization, comparative, negation, relational, inclusion, superlative, aggregational, rephrase or combinations of them** are non-trivial. But non-trvial is not limited to these types, any statement involving understanding and reasoning is accepted.

We list two examples below to help you understand, you are encouraged to open the **table wikipedia link** to understand the context of the table. (Everything in the table is lower-cased, you are free to use lower or upper case in your sentence):

Table Wikipedia Link: Road_Rules_Challenge:_The_Island
(https://en.wikipedia.org/wiki/Real_World/Road_Rules_Challenge:_The_Island)

| player | original season | gender | eliminated | placing |
|---|---|---|---|---|
| derrick kosinski | rr : x - treme | male | winner | winner |
| evelyn smith | fresh meat | female | winner | winner |
| johnny devenanzio | rw : key west | male | winner | winner |
| kenny santucci | fresh meat | male | winner | winner |
| jenn grijalva | rw : denver | female | episode 8 | runner - up |
| paula meronek | rw : key west | female | episode 8 | runner - up |
| robin hibbard | rw : san diego | female | episode 8 | runner - up |
| ryan kehoe | fresh meat | male | episode 8 | runner - up |
| dunbar merrill | rw : sydney | male | episode 8 | 9th place |
| johanna botta | rw : austin | female | episode 8 | 10th place |
| kellyanne judd | rw : sydney | female | episode 8 | 11th place |
| dan walsh | rr : viewers' revenge | male | episode 8 | 12th place |
| colie edison | rw : denver | female | episode 7 | 13th place |
| cohutta grindstaff | rw : sydney | male | episode 6 | 14th place |
| tyrie ballard | rw : denver | male | episode 5 | 15th place |
| ashli robson | rw : sydney | female | episode 4 | 16th place |
| rachel robinson | rr : campus crawl | female | episode 3 | 17th place |
| abram boise | rr : south pacific | male | episode 2 | 18th place |
| dave malinosky | rw : hollywood | male | episode 2 (quit) | 19th place |

**Rejected ("Trivial") examples**:
1. In the TV series "The Island", Derrick Kosinski is a male character. (Easy! You can simply look into first row to produce this sentence.)
2. Derrick Kosinski has the placing of winner in the TV series.
3. Kenny Santucci is from original season of "Fresh Meat".

4. Jenn Grijalva is Runner-Up of the challenge.

**Accepted ("Non-Trivial") examples**:
(Superlative): In the TV series "The Island", Evelyn Smith is the highest ranked female.
(Comparitive): In the TV series "The Island", Jenn Grijalva appears later than Colie Edison in the series.
(Relational): Ashli Robson appears one episode later than Rachel Robinson in the TV series.
(Summarization): there are three male winners in the challenge.
(Rephrase): Evelyn Smith never eliminated in any episode in the TV series.
(Combination): Derrick Kosinski is the winner and Jenn Grijalva is Runner-Up of the challenge.
(Negation): jenn grijalva is not the female winning the challenge.
(Inclusion): Evelyn smith is one of the four winner for the challenge.

Table Wikipedia Link: AFC_Champions_League (https://en.wikipedia.org/wiki/AFC_Champions_League)

| rank | member association | points | group stage | play - off | afc cup |
|------|--------------------|--------|-------------|------------|---------|
| 1 | saudi arabia | 860.5 | 4 | 0 | 0 |
| 2 | qatar | 838.2 | 4 | 0 | 0 |
| 3 | iran | 813.5 | 3 | 1 | 0 |
| 4 | uae | 750.2 | 2 | 2 | 0 |
| 5 | uzbekistan | 680.8 | 1 | 0 | 0 |
| 6 | india | 106.4 | 0 | 0 | 2 |
| 7 | jordan | 128.7 | 0 | 0 | 2 |

**Rejected ("Trivial") examples**:
1. In the rank, it has 0 play - off.
2. ratar is in rank 2.
3. When member association is india, the points is 106.4.

**Accepted ("Non-Trivial") examples**:
(Negation): iran is one of the two countries getting into the 4th stage. (Average): uae and qatar have an average of 1 play - off during the champion league.
(Algorithmic): saudi arabia achieves 22.3 more points than qatar.
(Comparison): india got lower points than jordan in the league.
(Summarization): there are two team which have won the afc cup twice.
(Superlative): In the Champions League, saudi arabia achieves the highest points.
(Combination): saudi arabia is the group stage 4 while iran is in group stage 3.

Tips1: We set minimum length to 9, and sentences with more complicated grammar structures are preferred.
Tips2: Do not limited to only one type of description like superlative or relative.
Tips3: Copying the records from the table is encouraged, which can help avoid typos and mis-spelling as much as possible, .
Tips4: Do not vague words like "maybe", "perhaps", "good", "excellent", "most", etc.

First Read the following table, then write five diverse **non-trivial** facts for this given table:

Table Source: athletics at the 1952 summer olympics - men 's pole vault
(https://en.wikipedia.org/wiki/Athletics_at_the_1952_Summer_Olympics_%E2%80%93_Men%27s_pole_vault)

| athlete | nationality | 3.60 | 3.80 | 3.95 | result |
|---|---|---|---|---|---|
| bob richards | united states | - | - | o | 4.55 or |
| don laz | united states | - | - | o | 4.50 |
| ragnar lundberg | sweden | - | - | o | 4.40 |
| petro denysenko | soviet union | - | - | o | 4.40 |
| valto olenius | finland | - | - | - | 4.30 |
| bunkichi sawada | japan | - | o | xxo | 4.20 |
| volodymyr brazhnyk | soviet union | - | o | o | 4.20 |
| viktor knyazev | soviet union | - | o | o | 4.20 |
| george mattos | united states | - | - | o | 4.20 |
| erkki kataja | finland | - | - | o | 4.10 |
| tamás homonnay | sweden | - | o | o | 4.10 |
| lennart lind | hungary | - | o | o | 4.10 |
| milan milakov | yugoslavia | - | o | xo | 4.10 |
| rigas efstathiadis | greece | - | o | o | 3.95 |
| torfy bryngeirsson | iceland | - | o | o | 3.95 |
| erling kaas | norway | - | o | xxx | 3.80 |
| theodosios balafas | greece | o | o | xxx | 3.80 |
| jukka piironen | finland | - | xo | xx | 3.80 |
| zeno dragomir | romania | - | xo | xx | 3.80 |

Please write a non-trivial statement, minimum 9 words

Please write a non-trivial statement, minimum 9 words

Please write a non-trivial statement, minimum 9 words

Please write a non-trivial statement, minimum 9 words

Please write a non-trivial statement, minimum 9 words

**Survey Instructions** (Click to expand)

Please first read a table to understand its content, an example is shown below, which contains the leaderboard of a competition.

| Player | Original Season | Gender | Eliminated | Placing |
|---|---|---|---|---|
| Derrick Kosinski | RR: X-Treme | Male | Winner | Winner |
| Evelyn Smith | Fresh Meat | Female | Winner | Winner |
| Johnny Devenanzio | RW: Key West | Male | Winner | Winner |
| Kenny Santucci | Fresh Meat | Male | Winner | Winner |
| Jenn Grijalva | RW: Denver | Female | Episode 8 | Runner-Up |
| Paula Meronek | RW: Key West | Female | Episode 8 | Runner-Up |
| Robin Hibbard | RW: San Diego | Female | Episode 8 | Runner-Up |
| Ryan Kehoe | Fresh Meat | Male | Episode 8 | Runner-Up |
| Dunbar Merrill | RW: Sydney | Male | Episode 8 | 9th Place |
| Johanna Botta | RW: Austin | Female | Episode 8 | 10th Place |
| KellyAnne Judd | RW: Sydney | Female | Episode 8 | 11th Place |
| Dan Walsh | RR: Viewers' Revenge | Male | Episode 8 | 12th Place |
| Colie Edison | RW: Denver | Female | Episode 7 | 13th Place |
| Cohutta Grindstaff | RW: Sydney | Male | Episode 6 | 14th Place |
| Tyrie Ballard | RW: Denver | Male | Episode 5 | 15th Place |
| Ashli Robson | RW: Sydney | Female | Episode 4 | 16th Place |
| Rachel Robinson | RR: Campus Crawl | Female | Episode 3 | 17th Place |
| Abram Boise | RR: South Pacific | Male | Episode 2 | 18th Place |
| Dave Malinosky | RW: Hollywood | Male | Episode 2 (quit) | 19th Place |
| Tonya Cooley | RW: Chicago | Female | Episode 1 | 20th Place |

You are given a sentence to describe a fact in the table, please follow the following two cases to finish the job:

**\* If the given sentence is fluent and consistent with the table, then please re-write it to make it "fake" based on the following criteria**:
1. Contradictory: it should still be a fluent and coherent, **but it needs be explicitly contrdictory to the facts in the table**.
2. Do not simply add **NOT** to revert the sentence meaning.
3. Do not write **neutral or non-verifiable sentences**, you need to confirm it in the table.
3. The fake statement needs to be clear, explicit and natural, do not use vague or ambiguous words like "bad", "good", "many", etc.
4. try to use diverse fake types during annotatoin.

Example 1. Given statement: Ashli Robson was eliminated in episode 4.
Good Faking: Ashli Robson survives through episode 1 to episode 5.
Good Faking: Ashli Robson is not the only one eliminated in episode 4.
Bad Faking (Simply add not): Ashli Robson was not eliminated on episode 4.

Bad Faking (Ambiguous, who is Ashli?): Ashli was not eliminated on episode 4.

Bad Faking (Irrelevant): Ashli was born in Mexico.

Bad Faking (Too subjective, what do you mean by "early"): AshlDerrick Kosinski lost the game very early.

Bad Faking (Not verifiable): AshlDerrick Kosinski was the most popular player.

Example 2. Given statement: Tonya Cooley is in the 20th place.

Good Faking: Tonya Cooley is not the last in placing.

Good Faking: Tonya Cooley is eliminated in episode 1 but not the last in placing.

Bad Faking: (There is nothing larger than 20th) Tonya Cooley is after the 20th place.

Bad Faking: (Half Wrong/half Right) When the gneder is female, the player is Tonya Colley.

Bad Faking (Introduce values outside the table): Tonya Cooley is in the 43th place.

Bad Faking (Typo): Tonya Cooler is in the 20th palace.

**\* If the given statement is erroneous (see following), please type in N/A in the input box.**

1. critical grammar error like missing verbs, nouns, etc. Do not count small errors like tense, singular/plural, case errors.

2. serious typo, misspelling.

3. the described fact is contradictory to the table.

**You can use the highlight button to help you find the mentions in the table, you can use either upper or lower case, not important**

First Read the given tables, then rewrite the statements to make them **fake**:

Table Source: 2003 - 04 isu junior grand prix
(https://en.wikipedia.org/wiki/2003%E2%80%9304_ISU_Junior_Grand_Prix)

| rank | nation | gold | silver | bronze | total |
|------|--------|------|--------|--------|-------|
| 1 | russia | 10 | 14 | 8 | 32 |
| 2 | united states | 9 | 6 | 7 | 22 |
| 3 | canada | 4 | 2 | 10 | 16 |
| 4 | japan | 4 | 5 | 4 | 13 |
| 5 | hungary | 4 | 0 | 2 | 6 |
| 6 | czech republic | 2 | 1 | 1 | 4 |
| 6 | ukraine | 1 | 3 | 0 | 4 |
| 6 | italy | 0 | 1 | 3 | 4 |
| 7 | sweden | 1 | 2 | 0 | 3 |
| 8 | israel | 1 | 1 | 0 | 2 |
| 9 | finland | 0 | 0 | 1 | 1 |
| 9 | france | 0 | 1 | 0 | 1 |

Hightlight Mentions, Click Me!

Given Statement: russia won the most silver medals in the grand prix

Please rewrite a sentence which is contradictory to the table

Hightlight Mentions, Click Me!

Given Statement: france and finland won the least medals in the grand prix

Please rewrite a sentence which is contradictory to the table

Hightlight Mentions, Click Me!

Given Statement: hungary and finland were the only countries that idd not win any silver medals

Please rewrite a sentence which is contradictory to the table

Hightlight Mentions, Click Me!

Given Statement: the united states won more gold medals than canada

Please rewrite a sentence which is contradictory to the table

Hightlight Mentions, Click Me!

Given Statement: canada won the most bronze medals in the grand prix

Please rewrite a sentence which is contradictory to the table

Submit

