# OpenReview forum: "TabFact: A Large-scale Dataset for Table-based Fact Verification"
_ICLR.cc/2020/Conference — Accept (Poster)_

### Official Review · AnonReviewer1 · 2019-10-16
**Official Blind Review #1**

**Rating:** 6

**Review:**

This paper proposes a new dataset for table-based fact verification and introduces a couple of methods for the task. I think that the dataset would be a useful resource (see some comments nevertheless on its construction), however the methods proposed are not particularly interesting, and the contributions to ML and NLP are overstated in my opinion. In addition the paper needs proof reading as there are many typos, some of which make comprehension problematic. In detail:

- The dataset is the main contribution of this paper. Its size is great. However I have some concerns on its construction. The guidelines described for constructing simple and complex claims are rathe vague, e.g. how does one define "too much symbolic reasoning" and explain it to crowdworkers? How was the linguistic complexity difference between the two channels measured? How were the claims sanity checked? In the beginning of section 2.3 it is stated that quality control filtered a substantial proportion of the statements. How was this done?

- A troublesome aspect in my opinion of the dataset is that it only allows for evaluation of the Entailed/Refuted binary classification task, but not on whether the model used the right evidence to reach its conclusion. Thus it will always be possible for models to score highly without doing the right thing. Avoiding trivial re-writes helps, but it is unlikely to be enough as human crowd workers will try to optimize their earnings.

- The two models discussed are OK as baselines, but not particularly interesting or appropriate. Both require substantial rule-based processing (named entity linking, latent program construction. templates) and eventually linearize structured data (the program or the table). I understand that this is not the main contribution of the paper, but given the substantial amount of work on semantic parsing and question answering I was expecting more appropriate baselines taking previous work into account. The performance of the model is not great either, especially if we consider that they are only evaluated on returning a binary label, not the correct evidence from the table.

- While it is true that most of the fact verification work has focus on textual sources, the challenge of combining reasoning over continuous and discrete representations is not new. The various QA works mentioned in the related work section address this, as well as work on theorem proving: https://arxiv.org/abs/1705.11040 Furthermore, there has been at least one more previous work on table based verification against  FreeBase tables: https://www.aclweb.org/anthology/D15-1312/. Thus I believe the discussion of the challenges posed by this dataset should be re-framed, especially given that the kinds of programs that need to be constructed are of similar complexity to previous work like the WikiTableQuestions.

- writing: "the model is expected to excel... but to fall short", "we follow the human subject research protocols" (which ones?), "in case of obvious stylistic patterns" (which ones). On the whole it is understandable, but the writing should be improved.

---- Post author response

I have responded to the author response and I have revised my score.

**Experience Assessment:**

I have published in this field for several years.

**Review Assessment: Checking Correctness Of Derivations And Theory:**

N/A

**Review Assessment: Checking Correctness Of Experiments:**

I carefully checked the experiments.

**Review Assessment: Thoroughness In Paper Reading:**

N/A

---

> ### Author Response · Authors · 2019-11-11
> **Author Response (Part 1)**
>
> 1. Guideline: The guideline for annotating complex claims are given in Appendix H, where we display the HIT interface for dataset collection. Since it was hard to explain these scientific terms to the crowdworkers, we made the following efforts to make sure the collected data follows our expectation: 1) we provide comprehensive examples for different logic operations to the annotators to help them understand what different operations represent. 2) we iterate a lot of rounds to refine our instruction to gradually improve annotator’s annotation skills. We reward good workers and encourage them to annotate more high-quality claims while penalizing low-quality workers. 3) our team employs 8 experts to perform the quality control during the annotation and after the annotation, where each expert spent over 50 hours on this project to ensure the quality of the dataset. We describe the details of the quality control step in Sec 2.2. The "too much reasoning ... " sentence is a bit vague, what we want to emphasize is to only involve unary facts in the statement rather than having compound facts within the statement, we revised it in the revision.
> 2. Linguistic/Logic Difference from two channels: The two channels have different instructions and examples to guide the crowd workers to behave differently. In the simple channel, our instruction explicitly encourages workers to annotate simple and straightforward claims about certain entities without strong paraphrasing and inference. To maximize the reward, the workers tend to annotate rather simple claims without too strong reasoning. In the complex channel, as depicted in the HIT interface in the Appendix, we explicitly enforce the workers to follow the examples to annotate claims with higher-order logic and linguistic paraphrase/inference. This channel involves more dedicated quality control to ensure quality.
> 3. Evaluation problem: we follow the standard NLI/Verification problems (e.g., SNLI, GLUE, SuperGLUE, FEVER) to use classification accuracy as one of our evaluation metrics. The binary classification gives a very objective and accurate evaluation of the end-task goal. To provide additional evaluation, in our original submission, we have already employed three additional metrics in Table 3 to give fine-grained analysis about different model components like entity linking/search/discriminator, etc.
> 4. Higher performance without learning well: We have already applied different state-of-the-art methods like BERT, but it still cannot achieve plausible accuracy and exhibits unstable training behavior. It indicates that the proposed new task is a challenging problem. To make sure the negative samples do not have explicit cues, we hired 8 experts to adopt strict quality control to enforce the workers to annotate high-quality and challenging negative sentences.
> 5. Previous semantic parsing baselines: to the best of our knowledge, the previous semantic parsing papers can be divided into two groups: 1) with logic form annotation: (Lu 2008, Artzi 2014, Dong 2016, Yin 2017, Zhong 2017, Cheng 2017, etc), these methods rely on annotated logic forms to learn the parsing model, which does not fit our case as we do not have logic-form annotation. 2) weakly-supervised: Traditional parsing models (Luke 2013, Liang 2013, Berant 2014, Wang 2015, Pasupat 2015) mostly rely on rules/lexicon/grammar and feature-based log-linear ranking models, etc. The core ideas of these methods are based on BFS/DFS/DP search with strong pruning strategies to harvest the plausible candidates and then score the most likely parses for answer prediction. Our LPA model inherits such a pipeline to first do BFS with memorization to accelerate the search speed, and then leverage a neural discriminator to rank the plausible parses.
> Recently, people have invented neural parsing models (Liang 2017, Rishabh 2019), which implements the parsing as a sequence generation architecture and leverages reinforcement learning to address the weakly supervised challenge. We have implemented NSM (Liang 2017) baseline under two settings, and we have updated the revision to include the experimental results in Table 2. Since we don’t have any annotated programs for bootstrapping, directly using RL to train NSM does not work well given the spurious program problem. By leveraging the LPA-searched programs to teacher-force NSM and then fine-tune it with RL can yield more reasonable performance. Through this comparison, we found that LPA actually performs better under our verification problem with under-specified rewards.
> We are not sure if I missed some publicly available methods, please remind us of that and we will try to implement it as a baseline for TabFact.

---

> ### Author Response · Authors · 2019-11-11
> **Author Response (Part 2)**
>
> 6. Relatedness to other QA datasets: We have updated the “related work” section to include more comprehensive comparisons with question answering/semantic parsing and other references suggested by the reviewers. Our proposed dataset is distinguished in the following aspects: 1) From the linguistic reasoning side, the existing QA dataset like WikitableQuestion is mostly dealing with surface-level linguistic phenomena like paraphrase/abbreviation/etc. In TabFact, since workers are making statements or conclusions, they tend to involve deeper linguistic reasoning like inference, commonsense. 2) From the symbolic side, the statement in TabFact tends to contain compound facts, all of them need to be verified to predict the verdict. It normally increases the complexity and length of the reasoning chain. These two major challenges have greatly distinguished TabFact from the known QA datasets and provide a new research direction for the community to explore.
> 7. We have corrected the minor issues in the revision, the minimum payment protocol is listed in https://en.wikipedia.org/wiki/Minimum_wage_in_the_United_States (cited in the revised version).
>
> [Lu 2008]: A generative model for parsing natural language to meaning representations
> [Artzi 2014]: Learning Compact Lexicons for CCG Semantic Parsing
> [Dong 2016]: Language to Logical Form with Neural Attention
> [Yin 2016]: A syntactic neural model for general-purpose code generation
> [Zhong 2017]: Seq2sql: Generating structured queries from natural language using reinforcement learning
> [Luke 2013]: Weakly supervised learning of semantic parsers for mapping instructions to actions
> [Liang 2013]: Learning dependency-based compositional semantics
> [Berant 2014]: Semantic parsing via paraphrasing
> [Wang 2015]: Building a semantic parser overnight.
> [Pasupat 2015]: Compositional semantic parsing on semi-structured tables.
> [Liang 2017]: Neural Symbolic Machines: Learning Semantic Parsers on Freebase with Weak Supervision
> [Rishabh 2019]: Learning to Generalize from Sparse and Underspecified Rewards

---

### Official Review · AnonReviewer2 · 2019-10-28
**Official Blind Review #2**

**Rating:** 6

**Review:**

This work proposes the problem of fact verification with semi-structured data source such as tables. Specifically, the authors created a new dataset TabFact and evaluated two baseline models with different variations. They applied two criteria and different rewards for workers to collect two subsets of different levels (“simple” and “complex”). They also applied a negative rewriting strategy to avoid exploitable cues or patterns in the annotations. They evaluated two baselines models: (1) latent program algorithm (LPA), which makes use of simple string match entity linking and systematic search and trains a neural network discriminator, and (2) Table-BERT, which linearize the table into a sequence through concatenation or template, and treat it as a classfication problem. Both showed reasonable accuracy (~68%), but still below human performance (92.1%) on a held out test set.

I would like to see the paper accepted because:
(1) it proposes an interesting task (table fact verification) with a clean dataset, and the experiments evaluated the ability of the current neural network models, such as BERT, or hybrid models, such as the LPA baseline, to perform (symbolic) reasoning;
(2) special care is done to ensure the dataset doesn't contain simple cues or patterns, which is a common pitfall in dataset collection, and the dataset is also validated through two reasonable baseline models.

Some weakness and concerns are:
(1) some error analysis of the baseline models are missing. For example, what types questions are hard/simple for table-BERT and what are hard/simple for LPA, and some comparison between them. This helps point out where the difficulty come from, for example, whether the difficulty is language understanding or symbolic reasoning.
(2) "To focus on statement verification against the table, we do not feed the caption to the model and simply mask the phrases in the statements which links to the caption with placeholders."
I am not sure this is the right thing to do since even the human annotators requires the caption to understand the context, why not also feed the caption into the model?
(3) Although the Figure 2 showed that the higher order operations are indeed used in a majority of questions, which measures the breadth of the reasoning, it is unclear about the depths of the required reasoning, i.e., how many operations / steps are required to achieve the correct answer. It would help if the average number of steps / operations required for answering the questions are shown.

If the above concerns are addressed, I will be willing to raise my score.


Minor comments:

The paper, especially the Appendix, requires some proof reading, for example, I believe the caption for Figure 5 in Appendix A is misplaced.

"... they are explicitly banned bring ..." -> "banned from bringing"

“As illustrated in Figure 6, the title only acts as a placeholder in the statements to make it sound more natural.” -> From the example, it seems the name of the player is kept unchanged in the sentence, which is different from a placeholder.

Suggestions:

I understand this might require some works, but it would be really helpful to add more comments and maybe examples for the functions shown in Figure 5 (you might need to split the table into two pages or have another table for examples), so that it can be used by works that follows the LPA approach.

“During the annotation, the workers are explicitly guided to modify the words, phrases or sentence structures but retain the sentence style/length to prevent from artificial cues”
Avoiding simple cues or patterns are important, so it will be good if more details can be shared (for example, the instructions/guidelines you showed to the workers).

==================================

Post rebuttal update:

Thanks for the update to the paper. I think this work provides a good dataset and some reasonable baselines, thus should be accepted.

However, I am still a bit concerned about the categorization of questions and the analysis on reasoning depth, because they are all based on the programs generated by LPA, while systematic search's recall is just 77% and the programs potentially contains a lot of spurious ones. So the categorization of the questions in Figure 11 and the number of reasoning steps in Figure 12 has to be taken with a grain of salt. It is worth annotating a few hundred, say 100-300 questions, manually with the ground truth programs for question distribution analysis or reasoning depth analysis, and confirm the numbers estimated using programs from LPA. Those manually annotated examples would also be a great addition to the dataset to aid the researchers for better analysis. A good example is the WikiTableQuestions dataset https://github.com/ppasupat/WikiTableQuestions/releases, which contains 300 manually annotated examples (annotated-all.examples), and they are used to analyze the distribution of the questions in the paper.

**Experience Assessment:**

I have published in this field for several years.

**Review Assessment: Checking Correctness Of Derivations And Theory:**

N/A

**Review Assessment: Checking Correctness Of Experiments:**

I assessed the sensibility of the experiments.

**Review Assessment: Thoroughness In Paper Reading:**

I read the paper thoroughly.

---

> ### Author Response · Authors · 2019-11-11
> **Author Response**
>
> Thanks for your constructive feedback, here are the answers to your questions:
> 1. Error Analysis: We have put the detailed error analysis in Appendix C (Error Analysis) in the revision. Specifically, we first qualitatively discussed the bottlenecks of LPA and BERT model. Then we held-out some data from the validation set to analyze the performance difference between the two models. It turns out that the linguistic inference case is much better handled by BERT while max/min/argmax/argmin/count cases are better handled by LPA.
> 2. Feeding Caption to model: During annotation, we have given special notice to the worker not to rewrite the part copied from the Wikipedia table title during the phase of annotation for refuted statements. It implies that the captions shouldn’t be regarded as evidence for verification. Thus, we masked them out in the experiments. However, we did do some auxiliary experiments to provide the caption as an additional input to the discriminator for the LPA method. We found that the performance is roughly the same without much change, we added the exact numbers to Table 2. For Table-BERT, the title is already given in the template linearization.
> 3. Reasoning Depth: We have added Appendix D (Reasoning Depth) in the revision, which aims to visualize the reasoning depth of LPA method. As can be seen from the figure therein, the reasoning steps are concentrated between 4-7 steps. It implies the difficulty and compositionality of the statements in TabFact.
> 4. Minor comments: 1) I have replaced the original table with a detailed version. Thanks for your reminder. 2) regarding the sentence “as illustrated in …”, I would like to clarify that the annotators can only rely on the title to gain background knowledge. They cannot rewrite information about the title to create a “refuted” statement.
> 5. Function Description: We added detailed function description and input/output examples in Appendix A and also list the function triggers we adopted to shrink the search space.
> 6. Instruction to workers: The detailed guidelines are added to Appendix H, which is the HIT interface we used for the workers to follow during annotation.

---

> ### Author Response · Authors · 2019-12-20
> **Thanks for your comment**
>
> Thanks a lot for your constructive feedback for the paper! we are collecting some ground truth programs for annotated statements, hopefully it's going to be released sometime in January. I think that will indeed help the existing models to improve their performances.

---

### Official Review · AnonReviewer4 · 2019-11-04
**Official Blind Review #4**

**Rating:** 8

**Review:**

Updated review:
Thank you for addressing the comments and making relevant edits. Additionally, the HIT section provides a lot more insights into the data collection process. I've updated my score based on the responses/edits made.

------

This paper is about a dataset (TABFACT) aimed at promoting research for fact-verification using semi-structured data as evidence. The paper highlights how the existing fact-verification studies have been restricted to work with unstructured evidence, and hence lack generalization to use-cases where the evidence is in a structured format (eg. databases). The paper also highlights how fact-verification with semi-structured evidence is challenging, since it involves both linguistic reasoning (for paraphrasing, entailment etc.) and symbolic reasoning (for operations like count, min, max etc.). To tackle this, the authors suggest two approaches as baselines on the dataset - one uses off-the-shelf BERT model for NLI; the other one focuses on symbolic reasoning and is based on program execution - which primarily uses lexical matching and a set of predefined operations (like count/max/min) to construct a program.

Apart from a few issues (mentioned below), the paper is well written. The authors have provided a detailed overview of their data collection/verification pipeline and related model/experiments. Overall, it seems like an interesting dataset and I'm inclined towards accepting the paper.

A few remarks/concerns are:
1. Usefulness of the dataset:
It seems limiting for a fact-verification dataset to restrict itself to a binary space i.e. entailed vs refuted. It is often the case, that statements are not completely true or false. For example, the 3rd refuted statement in Figure 1 is partially true (‘there are five candidates in total’). With a binary space for supervision, we don’t really know if the system is actually able to capture the linguistic and symbolic nuances present in the task. It is entirely possible for the system to “do well” without “learning well”, if the learning/output space is this coarse (as opposed to a dataset like Vlachos and Riedel, 2014).
2. Related work:
The paper talks about introducing a new ‘format’ of evidence (structured text) and talks about ‘unstructured text’ as the only ‘other’ format of evidence. It misses out on a highly related task that uses image as evidence (notable datasets being: CLEVR-Humans, NLVR/2, GQA). Either these should be included in the related work, or the authors should make it explicit that this work only deals with ‘textual’ evidence.
3. The dataset statistics in Table 1 don’t seem to add up (train+dev+test = (92,283 + 12,792 + 12,779) = 117,854 != 118,275 (=Total #Sentence)).
4. Page 3, Section 2.3: “we further perform quality control” -> a line or two to explain quality control?
5. Appendix C: No data/statistics have been provided to support the conclusion of the ablation study.

Minor remarks:
- Page 2: Section 2:
1. overtly -> overly
2. huge tables(e.g. -> huge tables (e.g.

- Page 3: Section 2.3
1. to filter 18% entailed of entailed statements -> to filter 18% entailed statements

- Page 4:
1. candidate) . we need to  -> candidate), we need to

**Experience Assessment:**

I have read many papers in this area.

**Review Assessment: Checking Correctness Of Derivations And Theory:**

N/A

**Review Assessment: Checking Correctness Of Experiments:**

I carefully checked the experiments.

**Review Assessment: Thoroughness In Paper Reading:**

I read the paper thoroughly.

---

> ### Author Response · Authors · 2019-11-11
> **Author Response**
>
> Thanks for your constructive feedback, here are the answers to your questions:
> 1. Usefulness of the dataset: we follow the standard NLI/Fact-Checking setting to divide the label into entailed/refuted, where a given claim with “any part of the claim containing misinformation” is viewed as refuted. In order to deal with the coarse space of output label, we propose to analyze the accuracy/F1/HITS@K score of entity linking/search/discriminator to further evaluate the fine-grained performance of the model. To encourage automatic evaluation, we plan to annotate a smaller amount of gold data for these components to enrich the evaluation metric.
> 2. Missing reference: I added some references and talked about their relatedness in the revision. Among the listed datasets, NLVR/2 is especially related to our task.
> 3. Mismatch Set size: We filtered roughly 400 sentences from some abnormal tables (which contains special tokens/hyperlink/mathematical formula, etc) after merging simple/complex channels and divide the 117,854 into train/val/test.
> 4. Quality Control: We added a Section 2.2 “quality control” to talk about the details of our quality control strategy, which is very critical during collection. We distribute the quality control workload to 8 different experts, where each one spent over 50 hours to ensure the data quality.
> 5. w/ vs. w/o caption ablation: We performed a user study before the collection, where we randomly match 50 pairs from two annotation tasks (w/ title and w/o title) and distribute them to our 8 experts to measure the language fluency and informativeness without telling them their sources. The user study indicates that the 8 experts consistently agree to favor the statements annotated with Wikipedia title/caption. The gap is significant and we immediately made the decision to provide workers with table titles as context.

---

### Public Comment · ~Matt_Gardner1 · 2019-10-16
**Why entailment over QA?**

Thanks, this looks like an interesting dataset!  I've been thinking about data formats recently, though, so I had some questions.  I'm not trying to give a critical comment here, just have a discussion about a topic that's on my mind.

You address this briefly in the related work, but I wondered if you could give some more discussion about the tradeoffs involved in entailment vs. QA as a format for this problem.  If your goal is to probe a machine's ability to understand table-structured data, as seems to be the case here, when should we use entailment, and when should we use QA?  I have published several papers on the WikiTableQuestions dataset; what are your arguments to me about why I should use TabFact instead (or in addition)?

I am also a bit surprised to see no mention of either NLVR or NLVR2 in here, as they seem incredibly related, just probing image understanding instead of table understanding.  The challenges involved in learning will be very similar.  One benefit of QA over NLI, as you use it here, is that QA gives you a much richer learning signal; with only binary supervision it is incredibly challenging to learn the compositional semantics of the statements, as there are many more spurious reasoning paths that lead to the correct answer.

What are the benefits of NLI over QA?  You mention that it's harder to do linking, but I'm not entirely convinced (I can write a question about "Peter and John" just as well as I can write a statement).  You mention that it's more natural to have more compositionality in a statement than a question.  Is this really true?  I could take all of your statements and pose them as yes/no questions.  If we allow those, I'm not sure what the difference is between NLI and QA, other than that I can also ask other things with QA, and I can't with NLI.  Additionally, forcing people to come up with negative examples in an NLI format (so you have some kind of balanced data) will tend to introduce data artifacts, where you don't have nearly so many problems with QA, which doesn't have a binary output that needs to be balanced.

I've thought about this quite a bit, but I'm coming from a very strong QA bias, and there are probably benefits of NLI that I'm missing.  Any help with enumerating them?

---

> ### Author Response · Authors · 2019-10-18
> **Thanks for your question**
>
> First of all, thank you so much for raising this insightful question. I didn't explain it in detail in the paper due to the page limit.
>
> I have read several QA papers from you and those papers have inspired me a lot. Here, I'm not suggesting that we should use entailment to replace QA. I'm proposing that we should probably consider more tasks to test model's ability to perform (formal) reasoning in addition to the existing QA (WikiTableQuestion Pasupat et al. 2015), Natural Language Instruction (CCG Yoav et. al 2013) tasks. The entailment task also seems to be a plausible testbed for such purpose and it also has its unique characteristics compared to the other two.
>
> For NLVR and NLVR2, I'm not quite familiar with the multi-modal literature, I will look into them and add those references in the future revision.
>
> Here I will list the difference between Fact Verification and QA.
> 1. From the task design perspective, the fact verification aims to evaluate the logic factualness of the given statement. Unlike the questions in QA task, the statement explicitly or implicitly contains both questions and answers. In order to verify the factualness, we need to extract both the questions and answers from the statement, which is actually non-trivial in most examples in TabFact. For example, given a statement "John is the only one out of the 16 players, who has won the gold medal more than five times", we need to break it down into the three QA pairs: (How many players are there?, 16), (Has John won gold medal more than five times? yes), (How many players have won the gold medals for more than five times, 1). All three sub-clauses need to hold to support the given statement. These three QA pairs have conjunctive relations as we do not know where the potential "error" could come from and we need to verify them one by one.  Other than conjunctive relations, we might also need to model disjunctive, negation relations between different clauses for some other statements. This difference makes entailment more difficult in certain aspects: 1. the verification model needs the decomposition capability to break down the statement into several QA pairs and understand their logical relations.  2. we need to verify multiple QA-pairs, where each QA-pair is viewed as a local semantic parse tree. These trees are connected through their logic relations to finally infer the verdict, thus the final semantic parse for the statement is a larger tree. That's why I claim that the verification task has more compositionality than standard QA task, but it seems to cause some confusion here. I will try to clarify this point more in the revision.
>
> 2. From the application perspective, fact verification is different from Question Answering. Fact verification aims to detect the misinformation from the web, social media, news or detect erroneous claims from documents and articles. The previous literature is mostly focused on using free-form text as evidence, Tabfact considers using the abundant structured world knowledge to enhance it. Furthermore, fact verification can also inspire methodology to enhance the factualness of machine-generated text. Assuming we need to deploy a chatbot for customer service, which can generate a response with deep neural networks. In order to make sure the replied messages are consistent with the backend knowledge, a fact verification model can be deployed to enhance its factualness.
>
> 3. An interesting observation is that some state-of-the-art black-box NLI methods like BIDAF, ESIM, BERT, ROBERTA, etc can be directly applied to Tabfact without using semantic parsing to achieve pretty reasonable accuracy. I didn't see such a phenomenon in the Table/KB QA tasks (I might miss some recent progress, correct me if you are aware of some). This indicates that the TabFact dataset can probably serve as a benchmark for both "formal semantics" models and the "deep reasoning" models.
>
> Finally, I would like to reiterate that QA is still the most popular task for evaluating model's capability to do formal grounding and logic reasoning. But in addition to that, we should consider diverse tasks to evaluate the performance of the existing reasoning models.

---

### Author Response · Authors · 2019-11-11
**Change Log**

Thanks a lot for the constructive feedback from the reviewers and the public. We have made some changes to the original version of our paper:
1. In the introduction section, we elaborate on the difference of our model with QA datasets in linguistic reasoning and symbolic reasoning aspects.
2. In the dataset section, we add a detailed description of our quality control step.
3. In the experiment section, we add NSM experimental results and "LPA with the table caption as additional input".
4. In related work, we add a more detailed comparison with QA dataset and visualize the two characteristics of our dataset.
5. In the appendix, we perform detailed error analysis about two models and provide the reasoning step distribution of LPA model.
6. In the appendix, we add the AMT HIT interface to demonstrate our instruction for annotation.

---

### Public Comment · ~Hsien_Yung_Peng1 · 2020-07-29
**Misspelling**

I think there are some spelling mistakes on your paper.
The last paragraph on page 4," bread-first-search ".

---

### Decision · Program_Chairs · 2019-12-19

**Decision:**

Accept (Poster)

**Comment:**

This paper presents a new dataset for fact verification in text from tables. The task is to identify whether a given claim is supported by the information presented in the table. The authors have also presented two baseline models, one based on BERT and based on symbolic reasoning which have an ok performance on the dataset but still very behind the human performance. The paper is well-written and the arguments and experiments presented in the paper are sound.

After reviewer comments, the authors have incorporated major changes in the paper. I recommend an Accept for the paper in its current form.